# SNIP: Single-shot Network Pruning based on Connection Sensitivity

**Namhoon Lee, Thalaiyasingam Ajanthan & Philip H. S. Torr**
University of Oxford
{namhoon,ajanthan,phst}@robots.ox.ac.uk

## Abstract

Pruning large neural networks while maintaining their performance is often desirable due to the reduced space and time complexity. In existing methods, pruning is done within an iterative optimization procedure with either heuristically designed pruning schedules or additional hyperparameters, undermining their utility. In this work, we present a new approach that prunes a given network once at initialization prior to training. To achieve this, we introduce a saliency criterion based on connection sensitivity that identifies structurally important connections in the network for the given task. This eliminates the need for both pretraining and the complex pruning schedule while making it robust to architecture variations. After pruning, the sparse network is trained in the standard way. Our method obtains extremely sparse networks with virtually the same accuracy as the reference network on the MNIST, CIFAR-10, and Tiny-ImageNet classification tasks and is broadly applicable to various architectures including convolutional, residual and recurrent networks. Unlike existing methods, our approach enables us to demonstrate that the retained connections are indeed relevant to the given task.

## 1 Introduction

Despite the success of deep neural networks in machine learning, they are often found to be highly overparametrized making them computationally expensive with excessive memory requirements. Pruning such large networks with minimal loss in performance is appealing for real-time applications, especially on resource-limited devices. In addition, compressed neural networks utilize the model capacity efficiently, and this interpretation can be used to derive better generalization bounds for neural networks (Arora et al. (2018)).

In network pruning, given a large reference neural network, the goal is to learn a much smaller subnetwork that mimics the performance of the reference network. The majority of existing methods in the literature attempt to find a subset of weights from the pretrained reference network either based on a saliency criterion (Mozer & Smolensky (1989); LeCun et al. (1990); Han et al. (2015)) or utilizing sparsity enforcing penalties (Chauvin (1989); Carreira-Perpiñán & Idelbayev (2018)). Unfortunately, since pruning is included as a part of an iterative optimization procedure, all these methods require many expensive *prune – retrain cycles* and heuristic design choices with additional hyperparameters, making them non-trivial to extend to new architectures and tasks.

In this work, we introduce a saliency criterion that identifies connections in the network that are important to the given task in a data-dependent way before training. Specifically, we discover important connections based on their influence on the loss function at a variance scaling initialization, which we call connection sensitivity. Given the desired sparsity level, redundant connections are pruned once prior to training (*i.e.*, single-shot), and then the sparse pruned network is trained in the standard way. Our approach has several attractive properties:

- *Simplicity.* Since the network is pruned once prior to training, there is no need for pretraining and complex pruning schedules. Our method has no additional hyperparameters and once pruned, training of the sparse network is performed in the standard way.
- *Versatility.* Since our saliency criterion chooses structurally important connections, it is robust to architecture variations. Therefore our method can be applied to various architectures including convolutional, residual and recurrent networks with no modifications.

- *Interpretability.* Our method determines important connections with a mini-batch of data at single-shot. By varying this mini-batch used for pruning, our method enables us to verify that the retained connections are indeed essential for the given task.

We evaluate our method on MNIST, CIFAR-10, and Tiny-ImageNet classification datasets with widely varying architectures. Despite being the simplest, our method obtains extremely sparse networks with virtually the same accuracy as the existing baselines across all tested architectures. Furthermore, we investigate the relevance of the retained connections as well as the effect of the network initialization and the dataset on the saliency score.

## 2 RELATED WORK

**Classical methods.** Essentially, early works in network pruning can be categorized into two groups (Reed (1993)): 1) those that utilize sparsity enforcing penalties; and 2) methods that prune the network based on some saliency criterion. The methods from the former category (Chauvin (1989); Weigend et al. (1991); Ishikawa (1996)) augment the loss function with some sparsity enforcing penalty terms (*e.g.*, $L_0$ or $L_1$ norm), so that back-propagation effectively penalizes the magnitude of the weights during training. Then weights below a certain threshold may be removed. On the other hand, classical saliency criteria include the sensitivity of the loss with respect to the neurons (Mozer & Smolensky (1989)) or the weights (Karnin (1990)) and Hessian of the loss with respect to the weights (LeCun et al. (1990); Hassibi et al. (1993)). Since these criteria are heavily dependent on the scale of the weights and are designed to be incorporated within the learning process, these methods are prohibitively slow requiring many iterations of pruning and learning steps. Our approach identifies redundant weights from an architectural point of view and prunes them once at the beginning before training.

**Modern advances.** In recent years, the increased space and time complexities as well as the risk of overfitting in deep neural networks prompted a surge of further investigation in network pruning. While Hessian based approaches employ the diagonal approximation due to its computational simplicity, impressive results (*i.e.*, extreme sparsity without loss in accuracy) are achieved using magnitude of the weights as the criterion (Han et al. (2015)). This made them the de facto standard method for network pruning and led to various implementations (Guo et al. (2016); Carreira-Perpiñán & Idelbayev (2018)). The magnitude criterion is also extended to recurrent neural networks (Narang et al. (2017)), yet with heavily tuned hyperparameter setting. Unlike our approach, the main drawbacks of magnitude based approaches are the reliance on pretraining and the expensive prune – retrain cycles. Furthermore, since pruning and learning steps are intertwined, they often require highly heuristic design choices which make them non-trivial to be extended to new architectures and different tasks. Meanwhile, Bayesian methods are also applied to network pruning (Ullrich et al. (2017); Molchanov et al. (2017a)) where the former extends the soft weight sharing in Nowlan & Hinton (1992) to obtain a sparse and compressed network, and the latter uses variational inference to learn the dropout rate which can then be used to prune the network. Unlike the above methods, our approach is simple and easily adaptable to any given architecture or task without modifying the pruning procedure.

**Network compression in general.** Apart from weight pruning, there are approaches focused on structured simplification such as pruning filters (Li et al. (2017); Molchanov et al. (2017b)), structured sparsity with regularizers (Wen et al. (2016)), low-rank approximation (Jaderberg et al. (2014)), matrix and tensor factorization (Novikov et al. (2015)), and sparsification using expander graphs (Prabhu et al. (2018)) or Erdős-Rényi random graph (Mocanu et al. (2018)). In addition, there is a large body of work on compressing the representation of weights. A non-exhaustive list includes quantization (Gong et al. (2014)), reduced precision (Gupta et al. (2015)) and binary weights (Hubara et al. (2016)). In this work, we focus on weight pruning that is free from structural constraints and amenable to further compression schemes.

## 3 NEURAL NETWORK PRUNING

The main hypothesis behind the neural network pruning literature is that neural networks are usually overparametrized, and comparable performance can be obtained by a much smaller network (Reed (1993)) while improving generalization (Arora et al. (2018)). To this end, the objective is to learn

a sparse network while maintaining the accuracy of the standard reference network. Let us first formulate neural network pruning as an optimization problem.

Given a dataset $\mathcal{D} = \{(\mathbf{x}_i, \mathbf{y}_i)\}_{i=1}^{n}$, and a desired sparsity level $\kappa$ (*i.e.*, the number of non-zero weights) neural network pruning can be written as the following constrained optimization problem:

$$\min_{\mathbf{w}} L(\mathbf{w}; \mathcal{D}) = \min_{\mathbf{w}} \frac{1}{n} \sum_{i=1}^{n} \ell(\mathbf{w}; (\mathbf{x}_i, \mathbf{y}_i)) , \qquad (1)$$

$$\text{s.t.} \quad \mathbf{w} \in \mathbb{R}^m, \quad \|\mathbf{w}\|_0 \leq \kappa .$$

Here, $\ell(\cdot)$ is the standard loss function (*e.g.*, cross-entropy loss), $\mathbf{w}$ is the set of parameters of the neural network, $m$ is the total number of parameters and $\| \cdot \|_0$ is the standard $L_0$ norm.

The conventional approach to optimize the above problem is by adding sparsity enforcing penalty terms (Chauvin (1989); Weigend et al. (1991); Ishikawa (1996)). Recently, Carreira-Perpiñán & Idelbayev (2018) attempts to minimize the above constrained optimization problem using the stochastic version of projected gradient descent (where the projection is accomplished by pruning). However, these methods often turn out to be inferior to saliency based methods in terms of resulting sparsity and require heavily tuned hyperparameter settings to obtain comparable results.

On the other hand, saliency based methods treat the above problem as selectively removing redundant parameters (or connections) in the neural network. In order to do so, one has to come up with a good criterion to identify such redundant connections. Popular criteria include magnitude of the weights, *i.e.*, weights below a certain threshold are redundant (Han et al. (2015); Guo et al. (2016)) and Hessian of the loss with respect to the weights, *i.e.*, the higher the value of Hessian, the higher the importance of the parameters (LeCun et al. (1990); Hassibi et al. (1993)), defined as follows:

$$s_j = \begin{cases} |w_j| , & \text{for magnitude based} \\ \frac{w_j^2 H_{jj}}{2} \quad \text{or} \quad \frac{w_j^2}{2 H_{jj}^{-1}} & \text{for Hessian based} . \end{cases} \qquad (2)$$

Here, for connection $j$, $s_j$ is the saliency score, $w_j$ is the weight, and $H_{jj}$ is the value of the Hessian matrix, where the Hessian $\mathbf{H} = \partial^2 L / \partial \mathbf{w}^2 \in \mathbb{R}^{m \times m}$. Considering Hessian based methods, the Hessian matrix is neither diagonal nor positive definite in general, approximate at best, and intractable to compute for large networks.

Despite being popular, both of these criteria depend on the scale of the weights and in turn require pretraining and are very sensitive to the architectural choices. For instance, different normalization layers affect the scale of the weights in a different way, and this would non-trivially affect the saliency score. Furthermore, pruning and the optimization steps are alternated many times throughout training, resulting in highly expensive *prune – retrain cycles*. Such an exorbitant requirement hinders the use of pruning methods in large-scale applications and raises questions about the credibility of the existing pruning criteria.

In this work, we design a criterion which directly measures the connection importance in a data-dependent manner. This alleviates the dependency on the weights and enables us to prune the network once at the beginning, and then the training can be performed on the sparse pruned network. Therefore, our method eliminates the need for the expensive prune – retrain cycles, and in theory, it can be an order of magnitude faster than the standard neural network training as it can be implemented using software libraries that support sparse matrix computations.

## 4 SINGLE-SHOT NETWORK PRUNING BASED ON CONNECTION SENSITIVITY

Given a neural network and a dataset, our goal is to design a method that can selectively prune redundant connections for the given task in a data-dependent way even before training. To this end, we first introduce a criterion to identify important connections and then discuss its benefits.

### 4.1 CONNECTION SENSITIVITY: ARCHITECTURAL PERSPECTIVE

Since we intend to measure the importance (or sensitivity) of each connection independently of its weight, we introduce auxiliary indicator variables $\mathbf{c} \in \{0, 1\}^m$ representing the connectivity of

parameters $\mathbf{w}$.[1] Now, given the sparsity level $\kappa$, Equation 1 can be correspondingly modified as:

$$\min_{\mathbf{c},\mathbf{w}} L(\mathbf{c} \odot \mathbf{w}; \mathcal{D}) = \min_{\mathbf{c},\mathbf{w}} \frac{1}{n} \sum_{i=1}^{n} \ell(\mathbf{c} \odot \mathbf{w}; (\mathbf{x}_i, \mathbf{y}_i)) \, , \tag{3}$$
$$\text{s.t.} \quad \mathbf{w} \in \mathbb{R}^m \, ,$$
$$\mathbf{c} \in \{0,1\}^m, \quad \|\mathbf{c}\|_0 \leq \kappa \, ,$$

where $\odot$ denotes the Hadamard product. Compared to Equation 1, we have doubled the number of learnable parameters in the network and directly optimizing the above problem is even more difficult. However, the idea here is that since we have separated the weight of the connection ($\mathbf{w}$) from whether the connection is present or not ($\mathbf{c}$), we may be able to determine the importance of each connection by measuring its effect on the loss function.

For instance, the value of $c_j$ indicates whether the connection $j$ is active ($c_j = 1$) in the network or pruned ($c_j = 0$). Therefore, to measure the effect of connection $j$ on the loss, one can try to measure the difference in loss when $c_j = 1$ and $c_j = 0$, keeping everything else constant. Precisely, the effect of removing connection $j$ can be measured by,

$$\Delta L_j(\mathbf{w}; \mathcal{D}) = L(\mathbf{1} \odot \mathbf{w}; \mathcal{D}) - L((\mathbf{1} - \mathbf{e}_j) \odot \mathbf{w}; \mathcal{D}) \, , \tag{4}$$

where $\mathbf{e}_j$ is the indicator vector of element $j$ (*i.e.*, zeros everywhere except at the index $j$ where it is one) and $\mathbf{1}$ is the vector of dimension $m$.

Note that computing $\Delta L_j$ for each $j \in \{1 \ldots m\}$ is prohibitively expensive as it requires $m + 1$ (usually in the order of millions) forward passes over the dataset. In fact, since $\mathbf{c}$ is binary, $L$ is not differentiable with respect to $\mathbf{c}$, and it is easy to see that $\Delta L_j$ attempts to measure the influence of connection $j$ on the loss function in this discrete setting. Therefore, by relaxing the binary constraint on the indicator variables $\mathbf{c}$, $\Delta L_j$ can be approximated by the derivative of $L$ with respect to $c_j$, which we denote $g_j(\mathbf{w}; \mathcal{D})$. Hence, the effect of connection $j$ on the loss can be written as:

$$\Delta L_j(\mathbf{w}; \mathcal{D}) \approx g_j(\mathbf{w}; \mathcal{D}) = \left. \frac{\partial L(\mathbf{c} \odot \mathbf{w}; \mathcal{D})}{\partial c_j} \right|_{\mathbf{c}=\mathbf{1}} = \lim_{\delta \to 0} \left. \frac{L(\mathbf{c} \odot \mathbf{w}; \mathcal{D}) - L((\mathbf{c} - \delta \, \mathbf{e}_j) \odot \mathbf{w}; \mathcal{D})}{\delta} \right|_{\mathbf{c}=\mathbf{1}} . \tag{5}$$

In fact, $\partial L/\partial c_j$ is an infinitesimal version of $\Delta L_j$, that measures the rate of change of $L$ with respect to an infinitesimal change in $c_j$ from $1 \to 1 - \delta$. This can be computed efficiently in one forward-backward pass using automatic differentiation, for all $j$ at once. Notice, this formulation can be viewed as perturbing the weight $w_j$ by a multiplicative factor $\delta$ and measuring the change in loss. This approximation is similar in spirit to Koh & Liang (2017) where they try to measure the influence of a datapoint to the loss function. Here we measure the influence of connections. Furthermore, $\partial L/\partial c_j$ is not to be confused with the gradient with respect to the weights ($\partial L/\partial w_j$), where the change in loss is measured with respect to an additive change in weight $w_j$.

Notably, our interest is to discover important (or sensitive) connections in the architecture, so that we can prune unimportant ones in single-shot, disentangling the pruning process from the iterative optimization cycles. To this end, we take the magnitude of the derivatives $g_j$ as the saliency criterion. Note that if the magnitude of the derivative is high (regardless of the sign), it essentially means that the connection $c_j$ has a considerable effect on the loss (either positive or negative), and it has to be preserved to allow learning on $w_j$. Based on this hypothesis, we define connection sensitivity as the normalized magnitude of the derivatives:

$$s_j = \frac{|g_j(\mathbf{w}; \mathcal{D})|}{\sum_{k=1}^{m} |g_k(\mathbf{w}; \mathcal{D})|} \, . \tag{6}$$

Once the sensitivity is computed, only the top-$\kappa$ connections are retained, where $\kappa$ denotes the desired number of non-zero weights. Precisely, the indicator variables $\mathbf{c}$ are set as follows:

$$c_j = \mathbb{1}[s_j - \tilde{s}_\kappa \geq 0] \, , \quad \forall j \in \{1 \ldots m\} \, , \tag{7}$$

where $\tilde{s}_\kappa$ is the $\kappa$-th largest element in the vector $\mathbf{s}$ and $\mathbb{1}[\cdot]$ is the indicator function. Here, for exactly $\kappa$ connections to be retained, ties can be broken arbitrarily.

We would like to clarify that the above criterion (Equation 6) is different from the criteria used in early works by Mozer & Smolensky (1989) or Karnin (1990) which do not entirely capture the

---

[1]Multiplicative coefficients (similar to $\mathbf{c}$) were also used for subset regression in Breiman (1995).

---

**Algorithm 1** SNIP: Single-shot Network Pruning based on Connection Sensitivity

---

**Require:** Loss function $L$, training dataset $\mathcal{D}$, sparsity level $\kappa$          ▷ Refer Equation 3
**Ensure:** $\|\mathbf{w}^*\|_0 \leq \kappa$
  1: $\mathbf{w} \leftarrow$ VarianceScalingInitialization          ▷ Refer Section 4.2
  2: $\mathcal{D}^b = \{(\mathbf{x}_i, \mathbf{y}_i)\}_{i=1}^b \sim \mathcal{D}$          ▷ Sample a mini-batch of training data
  3: $s_j \leftarrow \frac{\left|g_j(\mathbf{w};\mathcal{D}^b)\right|}{\sum_{k=1}^m \left|g_k(\mathbf{w};\mathcal{D}^b)\right|}, \quad \forall j \in \{1 \ldots m\}$          ▷ Connection sensitivity
  4: $\tilde{\mathbf{s}} \leftarrow$ SortDescending($\mathbf{s}$)
  5: $c_j \leftarrow \mathbb{1}[s_j - \tilde{s}_\kappa \geq 0], \quad \forall j \in \{1 \ldots m\}$          ▷ Pruning: choose top-$\kappa$ connections
  6: $\mathbf{w}^* \leftarrow \arg\min_{\mathbf{w} \in \mathbb{R}^m} L(\mathbf{c} \odot \mathbf{w}; \mathcal{D})$          ▷ Regular training
  7: $\mathbf{w}^* \leftarrow \mathbf{c} \odot \mathbf{w}^*$

---

connection sensitivity. The fundamental idea behind them is to identify elements (*e.g.* weights or neurons) that least degrade the performance when removed. This means that their saliency criteria (*i.e.* $-\partial L/\partial \mathbf{w}$ or $-\partial L/\partial \boldsymbol{\alpha}$; $\boldsymbol{\alpha}$ refers to the connectivity of neurons), in fact, depend on the loss value before pruning, which in turn, require the network to be pre-trained and iterative optimization cycles to ensure minimal loss in performance. They also suffer from the same drawbacks as the magnitude and Hessian based methods as discussed in Section 3. In contrast, our saliency criterion (Equation 6) is designed to measure the sensitivity as to how much influence elements have on the loss function regardless of whether it is positive or negative. This criterion alleviates the dependency on the value of the loss, eliminating the need for pre-training. These fundamental differences enable the network to be pruned at single-shot prior to training, which we discuss further in the next section.

## 4.2 SINGLE-SHOT PRUNING AT INITIALIZATION

Note that the saliency measure defined in Equation 6 depends on the value of weights $\mathbf{w}$ used to evaluate the derivative as well as the dataset $\mathcal{D}$ and the loss function $L$. In this section, we discuss the effect of each of them and show that it can be used to prune the network in single-shot with initial weights $\mathbf{w}$.

Firstly, in order to minimize the impact of weights on the derivatives $\partial L/\partial c_j$, we need to choose these weights carefully. For instance, if the weights are too large, the activations after the non-linear function (*e.g.*, sigmoid) will be saturated, which would result in uninformative gradients. Therefore, the weights should be within a sensible range. In particular, there is a body of work on neural network initialization (Goodfellow et al. (2016)) that ensures the gradients to be in a reasonable range, and our saliency measure can be used to prune neural networks at any such initialization.

Furthermore, we are interested in making our saliency measure robust to architecture variations. Note that initializing neural networks is a random process, typically done using normal distribution. However, if the initial weights have a fixed variance, the signal passing through each layer no longer guarantees to have the same variance, as noted by LeCun et al. (1998). This would make the gradient and in turn our saliency measure, to be dependent on the architectural characteristics. Thus, we advocate the use of variance scaling methods (*e.g.*, Glorot & Bengio (2010)) to initialize the weights, such that the variance remains the same throughout the network. By ensuring this, we empirically show that our saliency measure computed at initialization is robust to variations in the architecture.

Next, since the dataset and the loss function defines the task at hand, by relying on both of them, our saliency criterion in fact discovers the connections in the network that are important to the given task. However, the practitioner needs to make a choice on whether to use the whole training set, or a mini-batch or the validation set to compute the connection saliency. Moreover, in case there are memory limitations (*e.g.*, large model or dataset), one can accumulate the saliency measure over multiple batches or take an exponential moving average. In our experiments, we show that using only one mini-batch of a reasonable number of training examples can lead to effective pruning.

Finally, in contrast to the previous approaches, our criterion for finding redundant connections is simple and directly based on the sensitivity of the connections. This allows us to effectively identify and prune redundant connections in a single step even before training. Then, training can be performed on the resulting pruned (sparse) network. We name our method SNIP for Single-shot Network Pruning, and the complete algorithm is given in Algorithm 1.

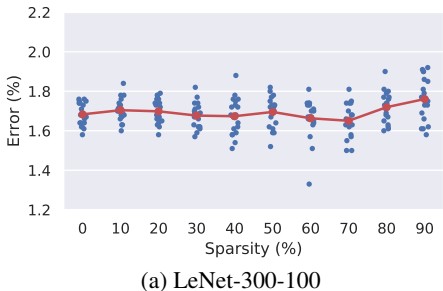 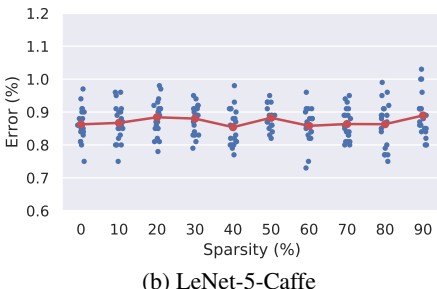

(a) LeNet-300-100          (b) LeNet-5-Caffe

Figure 1: Test errors of LeNets pruned at varying sparsity levels $\bar{\kappa}$, where $\bar{\kappa} = 0$ refers to the reference network trained without pruning. Our approach performs as good as the reference network across varying sparsity levels on both the models.

## 5 EXPERIMENTS

We evaluate our method, SNIP, on MNIST, CIFAR-10 and Tiny-ImageNet classification tasks with a variety of network architectures. Our results show that SNIP yields extremely sparse models with minimal or no loss in accuracy across all tested architectures, while being much simpler than other state-of-the-art alternatives. We also provide clear evidence that our method prunes genuinely explainable connections rather than performing blind pruning.

**Experiment setup** For brevity, we define the sparsity level to be $\bar{\kappa} = (m - \kappa)/m \cdot 100 \, (\%)$, where $m$ is the total number of parameters and $\kappa$ is the desired number of non-zero weights. For a given sparsity level $\bar{\kappa}$, the sensitivity scores are computed using a batch of $100$ and $128$ examples for MNIST and CIFAR experiments, respectively. After pruning, the pruned network is trained in the standard way. Specifically, we train the models using SGD with momentum of $0.9$, batch size of $100$ for MNIST and $128$ for CIFAR experiments and the weight decay rate of $0.0005$, unless stated otherwise. The initial learning rate is set to $0.1$ and decayed by $0.1$ at every 25k or 30k iterations for MNIST and CIFAR, respectively. Our algorithm requires no other hyperparameters or complex learning/pruning schedules as in most pruning algorithms. We spare $10\%$ of the training data as a validation set and used only $90\%$ for training. For CIFAR experiments, we use the standard data augmentation (*i.e.*, random horizontal flip and translation up to $4$ pixels) for both the reference and sparse models. The code can be found here: https://github.com/namhoonlee/snip-public.

### 5.1 PRUNING LENETS WITH VARYING LEVELS OF SPARSITY

We first test our approach on two standard networks for pruning, LeNet-300-100 and LeNet-5-Caffe. LeNet-300-100 consists of three fully-connected (fc) layers with 267k parameters and LeNet-5-Caffe consists of two convolutional (conv) layers and two fc layers with 431k parameters. We prune the LeNets for different sparsity levels $\bar{\kappa}$ and report the performance in error on the MNIST image classification task. We run the experiment 20 times for each $\bar{\kappa}$ by changing random seeds for dataset and network initialization. The results are reported in Figure 1.

The pruned sparse LeNet-300-100 achieves performances similar to the reference ($\bar{\kappa} = 0$), only with negligible loss at $\bar{\kappa} = 90$. For LeNet-5-Caffe, the performance degradation is nearly invisible. Note that our saliency measure does not require the network to be pre-trained and is computed at random initialization. Despite such simplicity, our approach prunes LeNets quickly (single-shot) and effectively (minimal accuracy loss) at varying sparsity levels.

### 5.2 COMPARISONS TO EXISTING APPROACHES

What happens if we increase the target sparsity to an extreme level? For example, would a model with only 1% of the total parameters still be trainable and perform well? We test our approach for extreme sparsity levels (*e.g.*, up to 99% sparsity on LeNet-5-Caffe) and compare with various pruning algorithms as follows: LWC (Han et al. (2015)), DNS (Guo et al. (2016)), LC (Carreira-Perpiñán &

| Method | Criterion | LeNet-300-100 | | LeNet-5-Caffe | | Pretrain | # Prune | Additional hyperparam. | Augment objective | Arch. constraints |
| --- | --- | --- | --- | --- | --- | --- | --- | --- | --- | --- |
| | | $\bar{\kappa}$ (%) | err. (%) | $\bar{\kappa}$ (%) | err. (%) | | | | | |
| Ref. | – | – | 1.7 | – | 0.9 | – | – | – | – | – |
| LWC | Magnitude | 91.7 | **1.6** | 91.7 | 0.8 | ✓ | many | ✓ | ✗ | ✓ |
| DNS | Magnitude | 98.2 | 2.0 | 99.1 | 0.9 | ✓ | many | ✓ | ✗ | ✓ |
| LC | Magnitude | 99.0 | 3.2 | 99.0 | 1.1 | ✓ | many | ✓ | ✓ | ✗ |
| SWS | Bayesian | 95.6 | 1.9 | 99.5 | 1.0 | ✓ | soft | ✓ | ✓ | ✗ |
| SVD | Bayesian | 98.5 | 1.9 | 99.6 | **0.8** | ✓ | soft | ✓ | ✓ | ✗ |
| OBD | Hessian | 92.0 | 2.0 | 92.0 | 2.7 | ✓ | many | ✓ | ✗ | ✗ |
| L-OBS | Hessian | 98.5 | 2.0 | 99.0 | 2.1 | ✓ | many | ✓ | ✗ | ✓ |
| SNIP (ours) | Connection sensitivity | 95.0 / 98.0 | **1.6** / 2.4 | 98.0 / 99.0 | **0.8** / 1.1 | ✗ | **1** | ✗ | ✗ | ✗ |

Table 1: Pruning results on LeNets and comparisons to other approaches. Here, "many" refers to an arbitrary number often in the order of total learning steps, and "soft" refers to soft pruning in Bayesian based methods. Our approach is capable of pruning up to 98% for LeNet-300-100 and 99% for LeNet-5-Caffe with marginal increases in error from the reference network. Notably, our approach is considerably simpler than other approaches, with no requirements such as pretraining, additional hyperparameters, augmented training objective or architecture dependent constraints.

Idelbayev (2018)), SWS (Ullrich et al. (2017)), SVD (Molchanov et al. (2017a)), OBD (LeCun et al. (1990)), L-OBS (Dong et al. (2017)). The results are summarized in Table 1.

We achieve errors that are comparable to the reference model, degrading approximately 0.7% and 0.3% while pruning 98% and 99% of the parameters in LeNet-300-100 and LeNet-5-Caffe respectively. For slightly relaxed sparsities (*i.e.*, 95% for LeNet-300-100 and 98% for LeNet-5-Caffe), the sparse models pruned by SNIP record better performances than the dense reference network. Considering 99% sparsity, our method efficiently finds 1% of the connections even before training, that are sufficient to learn as good as the reference network. Moreover, SNIP is competitive to other methods, yet it is unparalleled in terms of algorithm simplicity.

To be more specific, we enumerate some key points and non-trivial aspects of other algorithms and highlight the benefit of our approach. First of all, the aforementioned methods require networks to be fully trained (if not partly) before pruning. These approaches typically perform many pruning operations even if the network is well pretrained, and require additional hyperparameters (*e.g.*, pruning frequency in Guo et al. (2016), annealing schedule in Carreira-Perpiñán & Idelbayev (2018)). Some methods augment the training objective to handle pruning together with training, increasing the complexity of the algorithm (*e.g.*, augmented Lagrangian in Carreira-Perpiñán & Idelbayev (2018), variational inference in Molchanov et al. (2017a)). Furthermore, there are approaches designed to include architecture dependent constraints (*e.g.*, layer-wise pruning schemes in Dong et al. (2017)).

Compared to the above approaches, ours seems to cost almost nothing; it requires no pretraining or additional hyperparameters, and is applied only once at initialization. This means that one can easily plug-in SNIP as a preprocessor before training neural networks. Since SNIP prunes the network at the beginning, we could potentially expedite the training phase by training only the survived parameters (*e.g.*, reduced expected FLOPs in Louizos et al. (2018)). Notice that this is not possible for the aforementioned approaches as they obtain the maximum sparsity at the end of the process.

## 5.3 Various modern architectures

In this section we show that our approach is generally applicable to more complex modern network architectures including deep convolutional, residual and recurrent ones. Specifically, our method is applied to the following models:

- AlexNet-s and AlexNet-b: Models similar to Krizhevsky et al. (2012) in terms of the number of layers and size of kernels. We set the size of fc layers to 512 (AlexNet-s) and to 1024 (AlexNet-b) to adapt for CIFAR-10 and use strides of 2 for all conv layers instead of using pooling layers.
- VGG-C, VGG-D and VGG-like: Models similar to the original VGG models described in Simonyan & Zisserman (2015). VGG-like (Zagoruyko (2015)) is a popular variant adapted for CIFAR-10 which has one less fc layers. For all VGG models, we set the size of fc layers to 512, remove dropout layers to avoid any effect on sparsification and use batch normalization instead.
- WRN-16-8, WRN-16-10 and WRN-22-8: Same models as in Zagoruyko & Komodakis (2016).

| Architecture | Model | Sparsity (%) | # Parameters | Error (%) | Δ |
|---|---|---|---|---|---|
| Convolutional | AlexNet-s | 90.0 | 5.1m → 507k | 14.12 → 14.99 | +0.87 |
| | AlexNet-b | 90.0 | 8.5m → 849k | 13.92 → 14.50 | +0.58 |
| | VGG-C | 95.0 | 10.5m → 526k | 6.82 → 7.27 | +0.45 |
| | VGG-D | 95.0 | 15.2m → 762k | 6.76 → 7.09 | +0.33 |
| | VGG-like | 97.0 | 15.0m → 449k | 8.26 → 8.00 | **−0.26** |
| Residual | WRN-16-8 | 95.0 | 10.0m → 548k | 6.21 → 6.63 | +0.42 |
| | WRN-16-10 | 95.0 | 17.1m → 856k | 5.91 → 6.43 | +0.52 |
| | WRN-22-8 | 95.0 | 17.2m → 858k | 6.14 → 5.85 | **−0.29** |
| Recurrent | LSTM-s | 95.0 | 137k → 6.8k | 1.88 → 1.57 | **−0.31** |
| | LSTM-b | 95.0 | 535k → 26.8k | 1.15 → 1.35 | +0.20 |
| | GRU-s | 95.0 | 104k → 5.2k | 1.87 → 2.41 | +0.54 |
| | GRU-b | 95.0 | 404k → 20.2k | 1.71 → 1.52 | **−0.19** |

Table 2: Pruning results of the proposed approach on various modern architectures (before → after). AlexNets, VGGs and WRNs are evaluated on CIFAR-10, and LSTMs and GRUs are evaluated on the sequential MNIST classification task. The approach is generally applicable regardless of architecture types and models and results in a significant amount of reduction in the number of parameters with minimal or no loss in performance.

- LSTM-s, LSTM-b, GRU-s, GRU-b: One layer RNN networks with either LSTM (Zaremba et al. (2014)) or GRU (Cho et al. (2014)) cells. We develop two unit sizes for each cell type, 128 and 256 for $\{\cdot\}$-s and $\{\cdot\}$-b, respectively. The model is adapted for the sequential MNIST classification task, similar to Le et al. (2015). Instead of processing pixel-by-pixel, however, we perform row-by-row processing (*i.e.*, the RNN cell receives each row at a time).

The results are summarized in Table 2. Overall, our approach prunes a substantial amount of parameters in a variety of network models with minimal or no loss in accuracy ($< 1\%$). Our pruning procedure does not need to be modified for specific architectural variations (*e.g.*, recurrent connections), indicating that it is indeed versatile and scalable. Note that prior art that use a saliency criterion based on the weights (*i.e.*, magnitude or Hessian based) would require considerable adjustments in their pruning schedules as per changes in the model.

We note of a few challenges in directly comparing against others: different network specifications, learning policies, datasets and tasks. Nonetheless, we provide a few comparison points that we found in the literature. On CIFAR-10, SVD prunes $97.9\%$ of the connections in VGG-like with no loss in accuracy (ours: $97\%$ sparsity) while SWS obtained $93.4\%$ sparsity on WRN-16-4 but with a non-negligible loss in accuracy of $2\%$. There are a couple of works attempting to prune RNNs (*e.g.*, GRU in Narang et al. (2017) and LSTM in See et al. (2016)). Even though these methods are specifically designed for RNNs, none of them are able to obtain extreme sparsity without substantial loss in accuracy reflecting the challenges of pruning RNNs. To the best of our knowledge, we are the first to demonstrate on convolutional, residual and recurrent networks for extreme sparsities without requiring additional hyperparameters or modifying the pruning procedure.

## 5.4 UNDERSTANDING WHICH CONNECTIONS ARE BEING PRUNED

So far we have shown that our approach can prune a variety of deep neural network architectures for extreme sparsities without losing much on accuracy. However, it is not clear yet which connections are actually being pruned away or whether we are pruning the right (*i.e.*, unimportant) ones. What if we could actually peep through our approach into this inspection?

Consider the first layer in LeNet-300-100 parameterized by $\mathbf{w}_{l=1} \in \mathbb{R}^{784 \times 300}$. This is a layer fully connected to the input where input images are of size $28 \times 28 = 784$. In order to understand which connections are retained, we can visualize the binary connectivity mask for this layer $\mathbf{c}_{l=1}$, by averaging across columns and then reshaping the vector into 2D matrix (*i.e.*, $\mathbf{c}_{l=1} \in \{0,1\}^{784 \times 300} \rightarrow \mathbb{R}^{784} \rightarrow \mathbb{R}^{28 \times 28}$). Recall that our method computes $\mathbf{c}$ using a mini-batch of examples. In this experiment, we curate the mini-batch of examples of the same class and see which weights are retained for that mini-batch of data. We repeat this experiment for all classes

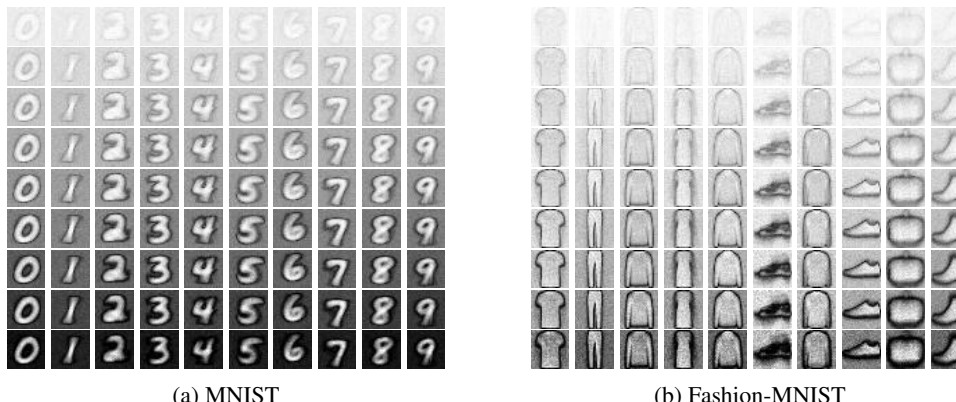

(a) MNIST (b) Fashion-MNIST

Figure 2: Visualizations of pruned parameters of the first layer in LeNet-300-100; the parameters are reshaped to be visualized as an image. Each column represents the visualizations for a particular class obtained using a batch of 100 examples with varying levels of sparsity $\bar{\kappa}$, from 10 (top) to 90 (bottom). Bright pixels indicate that the parameters connected to these region had high importance scores ($\mathbf{s}$) and survived from pruning. As the sparsity increases, the parameters connected to the discriminative part of the image for classification survive and the irrelevant parts get pruned.

(*i.e.*, digits for MNIST and fashion items for Fashion-MNIST) with varying sparsity levels $\bar{\kappa}$. The results are displayed in Figure 2 (see Appendix A for more results).

The results are significant; important connections seem to reconstruct either the complete image (MNIST) or silhouettes (Fashion-MNIST) of input class. When we use a batch of examples of the digit 0 (*i.e.*, the first column of MNIST results), for example, the parameters connected to the foreground of the digit 0 survive from pruning while the majority of background is removed. Also, one can easily determine the identity of items from Fashion-MNIST results. This clearly indicates that our method indeed prunes the *unimportant* connections in performing the classification task, receiving signals only from the most discriminative part of the input. This stands in stark contrast to other pruning methods from which carrying out such inspection is not straightforward.

## 5.5 EFFECTS OF DATA AND WEIGHT INITIALIZATION

Recall that our connection saliency measure depends on the network weights $\mathbf{w}$ as well as the given data $\mathcal{D}$ (Section 4.2). We study the effect of each of these in this section.

**Effect of data.** Our connection saliency measure depends on a mini-batch of train examples $\mathcal{D}^b$ (see Algorithm 1). To study the effect of data, we vary the batch size used to compute the saliency ($|\mathcal{D}^b|$) and check which connections are being pruned as well as how much performance change this results in on the corresponding sparse network. We test with LeNet-300-100 to visualize the remaining parameters, and set the sparsity level $\bar{\kappa} = 90$. Note that the batch size used for training remains the same as 100 for all cases. The results are displayed in Figure 3.

**Effect of initialization.** Our approach prunes a network at a stochastic initialization as discussed. We study the effect of the following initialization methods: 1) RN (random normal), 2) TN (truncated random normal), 3) VS-X (a variance scaling method using Glorot & Bengio (2010)), and 4) VS-H (a variance scaling method He et al. (2015)). We test on LeNets and RNNs on MNIST and run 20 sets of experiments by varying the seed for initialization. We set the sparsity level $\bar{\kappa} = 90$, and train with Adam optimizer (Kingma & Ba (2015)) with learning rate of 0.001 without weight decay. Note that for training VS-X initialization is used in all the cases. The results are reported in Figure 3.

For all models, VS-H achieves the best performance. The differences between initializers are marginal on LeNets, however, variance scaling methods indeed turns out to be essential for complex RNN models. This effect is significant especially for GRU where without variance scaling initialization, the pruned networks are unable to achieve good accuracies, even with different optimizers. Overall, initializing with a variance scaling method seems crucial to making our saliency measure reliable and model-agnostic.

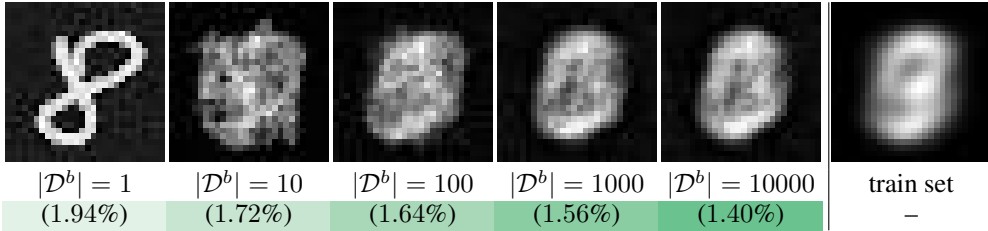

| $|\mathcal{D}^b| = 1$ | $|\mathcal{D}^b| = 10$ | $|\mathcal{D}^b| = 100$ | $|\mathcal{D}^b| = 1000$ | $|\mathcal{D}^b| = 10000$ | train set |
|---|---|---|---|---|---|
| (1.94%) | (1.72%) | (1.64%) | (1.56%) | (1.40%) | – |

Figure 3: The effect of different batch sizes: (top-row) survived parameters in the first layer of LeNet-300-100 from pruning visualized as images; (bottom-row) the performance in errors of the pruned networks. For $|\mathcal{D}^b| = 1$, the sampled example was $8$; our pruning precisely retains the valid connections. As $|\mathcal{D}^b|$ increases, survived parameters get close to the average of all examples in the train set (last column), and the error decreases.

| Init. | LeNet-300-100 | LeNet-5-Caffe | LSTM-s | GRU-s |
|---|---|---|---|---|
| RN | $1.90 \pm (0.09)$ | $0.89 \pm (0.04)$ | $2.93 \pm (0.20)$ | $47.61 \pm (20.49)$ |
| TN | $1.96 \pm (0.11)$ | $0.87 \pm (0.05)$ | $3.03 \pm (0.17)$ | $46.48 \pm (22.25)$ |
| VS-X | $1.91 \pm (0.10)$ | $0.88 \pm (0.07)$ | $1.48 \pm (0.09)$ | $\mathbf{1.80} \pm (0.10)$ |
| VS-H | $\mathbf{1.88} \pm (0.10)$ | $\mathbf{0.85} \pm (0.05)$ | $\mathbf{1.47} \pm (0.08)$ | $\mathbf{1.80} \pm (0.14)$ |

Table 3: The effect of initialization on our saliency score. We report the classification errors ($\pm$std). Variance scaling initialization (VS-X, VS-H) improves the performance, especially for RNNs.

## 5.6 FITTING RANDOM LABELS

To further explore the use cases of SNIP, we run the experiment introduced in Zhang et al. (2017) and check whether the sparse network obtained by SNIP memorizes the dataset. Specifically, we train LeNet-5-Caffe for both the reference model and pruned model (with $\bar{\kappa} = 99$) on MNIST with either true or randomly shuffled labels. To compute the connection sensitivity, always true labels are used. The results are plotted in Figure 4.

Given true labels, both the reference (red) and pruned (blue) models quickly reach to almost zero training loss. However, the reference model provided with random labels (green) also reaches to very low training loss, even

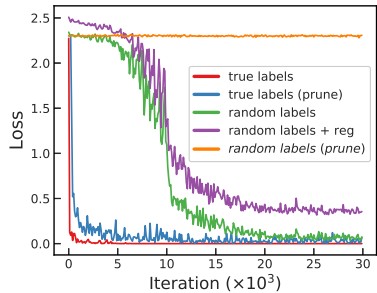

Figure 4: The sparse model pruned by SNIP does not fit the random labels.

with an explicit L2 regularizer (purple), indicating that neural networks have enough capacity to memorize completely random data. In contrast, the model pruned by SNIP (orange) fails to fit the random labels (high training error). The potential explanation is that the pruned network does not have sufficient capacity to fit the random labels, but it is able to classify MNIST with true labels, reinforcing the significance of our saliency criterion. It is possible that a similar experiment can be done with other pruning methods (Molchanov et al. (2017a)), however, being simple, SNIP enables such exploration much easier. We provide a further analysis on the effect of varying $\bar{\kappa}$ in Appendix B.

## 6 DISCUSSION AND FUTURE WORK

In this work, we have presented a new approach, SNIP, that is simple, versatile and interpretable; it prunes irrelevant connections for a given task at single-shot prior to training and is applicable to a variety of neural network models without modifications. While SNIP results in extremely sparse models, we find that our connection sensitivity measure itself is noteworthy in that it diagnoses important connections in the network from a purely untrained network. We believe that this opens up new possibilities beyond pruning in the topics of understanding of neural network architectures, multi-task transfer learning and structural regularization, to name a few. In addition to these potential directions, we intend to explore the generalization capabilities of sparse networks.

ACKNOWLEDGEMENTS

This work was supported by the Korean Government Graduate Scholarship, the ERC grant ERC-2012-AdG 321162-HELIOS, EPSRC grant Seebibyte EP/M013774/1 and EPSRC/MURI grant EP/N019474/1. We would also like to acknowledge the Royal Academy of Engineering and FiveAI.

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

## A  VISUALIZING PRUNED PARAMETERS ON (INVERTED) (FASHION-)MNIST

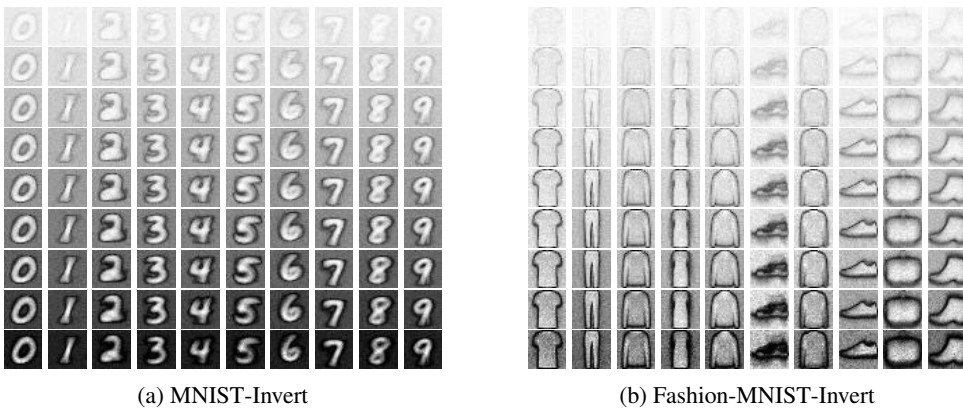

(a) MNIST-Invert      (b) Fashion-MNIST-Invert

Figure 5: Results of pruning with SNIP on inverted (Fashion-)MNIST (*i.e.*, dark and bright regions are swapped). Notably, even if the data is inverted, the results are the same as the ones on the original (Fashion-)MNIST in Figure 2.

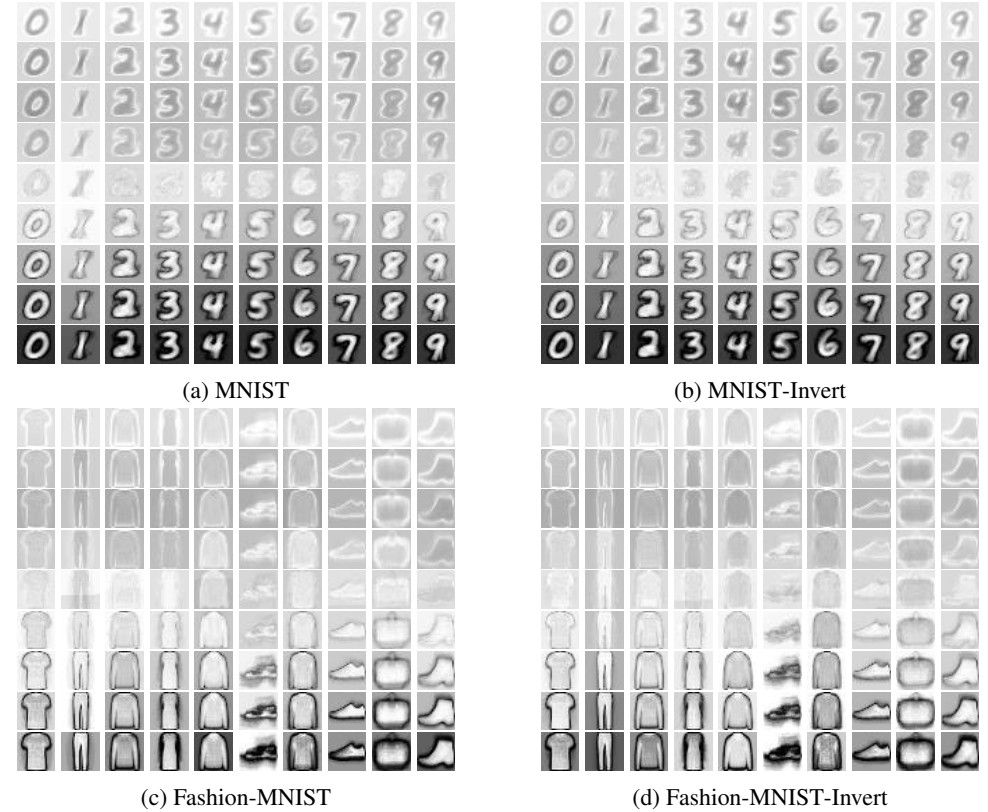

(a) MNIST      (b) MNIST-Invert

(c) Fashion-MNIST      (d) Fashion-MNIST-Invert

Figure 6: Results of pruning with $\partial L/\partial \mathbf{w}$ on the original and inverted (Fashion-)MNIST. Notably, compared to the case of using SNIP (Figures 2 and 5), the results are different: Firstly, the results on the original (Fashion-)MNIST (*i.e.*, (a) and (c) above) are not the same as the ones using SNIP (*i.e.*, (a) and (b) in Figure 2). Moreover, the pruning patterns are inconsistent with different sparsity levels, either intra-class or inter-class. Furthermore, using $\partial L/\partial \mathbf{w}$ results in different pruning patterns between the original and inverted data in some cases (*e.g.*, the 2nd columns between (c) and (d)).

# B  FITTING RANDOM LABELS: VARYING SPARSITY LEVELS

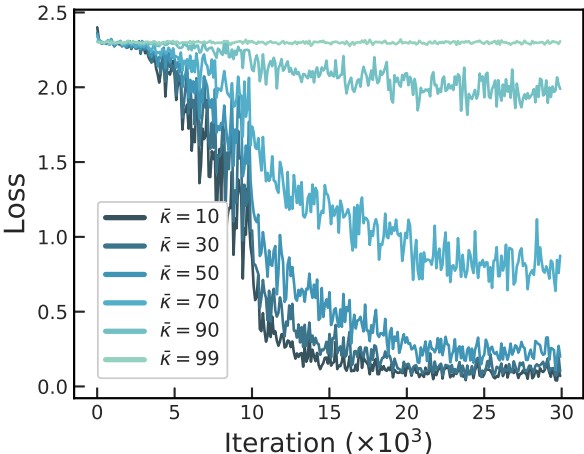

Figure 7: The effect of varying sparsity levels ($\bar{\kappa}$). The lower $\bar{\kappa}$ becomes, the lower training loss is recorded, meaning that a network with more parameters is more vulnerable to fitting random labels. Recall, however, that all pruned models are able to learn to perform the classification task without losing much accuracy (see Figure 1). This potentially indicates that the pruned network does not have sufficient capacity to fit the random labels, but it is capable of performing the classification.

# C  TINY-IMAGENET

| Architecture | Model | Sparsity (%) | # Parameters | Error (%) | $\Delta$ |
|---|---|---|---|---|---|
| | AlexNet-s | 90.0 | 5.1m $\rightarrow$ 507k | 62.52 $\rightarrow$ 65.27 | +2.75 |
| | AlexNet-b | 90.0 | 8.5m $\rightarrow$ 849k | 62.76 $\rightarrow$ 65.54 | +2.78 |
| Convolutional | VGG-C | 95.0 | 10.5m $\rightarrow$ 526k | 56.49 $\rightarrow$ 57.48 | +0.99 |
| | VGG-D | 95.0 | 15.2m $\rightarrow$ 762k | 56.85 $\rightarrow$ 57.00 | +0.15 |
| | VGG-like | 95.0 | 15.0m $\rightarrow$ 749k | 54.86 $\rightarrow$ 55.73 | +0.87 |

Table 4: Pruning results of SNIP on Tiny-ImageNet (before $\rightarrow$ after). Tiny-ImageNet[2] is a subset of the full ImageNet: there are 200 classes in total, each class has 500 and 50 images for training and validation respectively, and each image has the spatial resolution of $64 \times 64$. Compared to CIFAR-10, the resolution is doubled, and to deal with this, the stride of the first convolution in all architectures is doubled, following the standard practice for this dataset. In general, the Tiny-ImageNet classification task is considered much more complex than MNIST or CIFAR-10. Even on Tiny-ImageNet, however, SNIP is still able to prune a large amount of parameters with minimal loss in performance. AlexNet models lose more accuracies than VGGs, which may be attributed to the fact that the first convolution stride for AlexNet is set to be $4$ (by its design of no pooling) which is too large and could lead to high loss of information when pruned.

---

[2]https://tiny-imagenet.herokuapp.com/

# D  ARCHITECTURE DETAILS

| Module | Weight | Stride | Bias | BatchNorm | ReLU |
|--------|--------|--------|------|-----------|------|
| Conv | $[11, 11, 3, 96]$ | $[2, 2]$ | $[96]$ | ✓ | ✓ |
| Conv | $[5, 5, 96, 256]$ | $[2, 2]$ | $[256]$ | ✓ | ✓ |
| Conv | $[3, 3, 256, 384]$ | $[2, 2]$ | $[384]$ | ✓ | ✓ |
| Conv | $[3, 3, 384, 384]$ | $[2, 2]$ | $[384]$ | ✓ | ✓ |
| Conv | $[3, 3, 384, 256]$ | $[2, 2]$ | $[256]$ | ✓ | ✓ |
| Linear | $[256, 1024 \times k]$ | – | $[1024 \times k]$ | ✓ | ✓ |
| Linear | $[1024 \times k, 1024 \times k]$ | – | $[1024 \times k]$ | ✓ | ✓ |
| Linear | $[1024 \times k, c]$ | – | $[c]$ | ✗ | ✗ |

Table 5: AlexNet-s ($k = 1$) and AlexNet-b ($k = 2$). In the last layer, $c$ denotes the number of possible classes: $c = 10$ for CIFAR-10 and $c = 200$ for Tiny-ImageNet. The strides in the first convolution layer for Tiny-ImageNet are set $[4, 4]$ instead of $[2, 2]$ to deal with the increase in the image resolution.

| Module | Weight | Stride | Bias | BatchNorm | ReLU |
|--------|--------|--------|------|-----------|------|
| Conv | $[3, 3, 3, 64]$ | $[1, 1]$ | $[64]$ | ✓ | ✓ |
| Conv | $[3, 3, 64, 64]$ | $[1, 1]$ | $[64]$ | ✓ | ✓ |
| Pool | – | $[2, 2]$ | – | ✗ | ✗ |
| Conv | $[3, 3, 64, 128]$ | $[1, 1]$ | $[128]$ | ✓ | ✓ |
| Conv | $[3, 3, 128, 128]$ | $[1, 1]$ | $[128]$ | ✓ | ✓ |
| Pool | – | $[2, 2]$ | – | ✗ | ✗ |
| Conv | $[3, 3, 128, 256]$ | $[1, 1]$ | $[256]$ | ✓ | ✓ |
| Conv | $[3, 3, 256, 256]$ | $[1, 1]$ | $[256]$ | ✓ | ✓ |
| Conv | $[1/3/3, 1/3/3, 256, 256]$ | $[1, 1]$ | $[256]$ | ✓ | ✓ |
| Pool | – | $[2, 2]$ | – | ✗ | ✗ |
| Conv | $[3, 3, 256, 512]$ | $[1, 1]$ | $[512]$ | ✓ | ✓ |
| Conv | $[3, 3, 512, 512]$ | $[1, 1]$ | $[512]$ | ✓ | ✓ |
| Conv | $[1/3/3, 1/3/3, 512, 512]$ | $[1, 1]$ | $[512]$ | ✓ | ✓ |
| Pool | – | $[2, 2]$ | – | ✗ | ✗ |
| Conv | $[3, 3, 512, 512]$ | $[1, 1]$ | $[512]$ | ✓ | ✓ |
| Conv | $[3, 3, 512, 512]$ | $[1, 1]$ | $[512]$ | ✓ | ✓ |
| Conv | $[1/3/3, 1/3/3, 512, 512]$ | $[1, 1]$ | $[512]$ | ✓ | ✓ |
| Pool | – | $[2, 2]$ | – | ✗ | ✗ |
| Linear | $[512, 512]$ | – | $[512]$ | ✓ | ✓ |
| Linear | $[512, 512]$ | – | $[512]$ | ✓ | ✓ |
| Linear | $[512, c]$ | – | $[c]$ | ✗ | ✗ |

Table 6: VGG-C/D/like. In the last layer, $c$ denotes the number of possible classes: $c = 10$ for CIFAR-10 and $c = 200$ for Tiny-ImageNet. The strides in the first convolution layer for Tiny-ImageNet are set $[2, 2]$ instead of $[1, 1]$ to deal with the increase in the image resolution. The second Linear layer is only used in VGG-C/D.

