# OpenReview forum: "SNIP: SINGLE-SHOT NETWORK PRUNING BASED ON CONNECTION SENSITIVITY"
_ICLR.cc/2019/Conference_

### Official Review · AnonReviewer3 · 2018-10-31
**An interesting paper**

**Rating:** 9
**Confidence:** 4

**Review:**

Summary
The paper focuses on pruning neural networks. They propose to identify the nodes to be pruned even before training the whole network (conventionally, it is done as a separate step after the nn was trained and involves a number of iterations of retraining pruned nn). This initial step that identifies the connections to be pruned works off a mini-batch of data.

Authors introduce  a criterion to be used for identifying important parts of the network (connection sensitivity), that does not depend on the magnitude of the weights for neurons: they start by introducing a set of binary weights (one per a weight from a neuron) that indicate whether the connection is on or off and can be removed. Reformulating the optimization problem and relaxing the constraints on the binary weights, they approximate the sensitivity of the loss with respect to these indicator variables via the gradient. Then the normalized magnitude of these gradients is used to chose the connections to keep (keeping top k connections)

Clarity:
Well written, easy to follow

Detailed comments
Overall, very interesting. Seemingly very simple idea that seem to work well.
Table 2 does look impressive and it seems that it also reduces the overfiting, and the experiment with random labels on mnist seem to demonstrate that the method indeed preserves only connections relevant to the real labels, simplifying the architecture to a point when it cant just memorize the data

Several questions/critiques:
- When you relax the binary constraints, it becomes an approximation to an optimization problem, any indication of how far you are off solving it this way?
- For the initialization method of the weights, you seem to state that VS-H is the one to use. I wonder if it actually task dependent and architecture dependent. If yes, then the propose method still has a hyperparameter - how to initialize the weights initially
- How does it compare with just randomly dropping the connections or dropping them based on the magnitude of the initial weights.  It seems that the meat comes from the fact that you are able to use the label and good initial values, i wonder if just doing a couple of iterations of forward-backprop and then dropping the weights based on their magnitude can give you comparable results
- How does it compare to a distillation - it does not involve many cycles of retraining and can speed up inference time too
-Can it replace the architecture search - initialize a large architecture, use the method to prune the connections and here you go. Did you try that instead of using already pre-tuned architectures like AlexNet.

---

> ### Author Response · Authors · 2018-11-21
> **Response to R3**
>
> Thank you for the interest in our work and positive feedback. We find the comments highly insightful and address the key points below.
>
> # Optimizing c
> - We have attempted to optimize c and w together in an alternating optimization paradigm. Specifically, at each iteration, we fix one variable and optimize the other, and vice versa. We were able to achieve sparse networks with comparable accuracies to the reference network in some cases, however, in general the optimization was quite unstable. We believe that this is a promising direction to pursue, and yet further investigation will be required.
>
> # VS-H singularity and its dependency on task or architecture
> - We have further tested several variance scaling methods (including VS-X and VS-H) with different hyperparameters (e.g. distribution type and fan mode) and observed that all variance scaling initialization methods are robust to various architectures and models used in our work. It would be interesting to see how it behaves on different tasks other than the image classification task, and we are keen on exploring more on this as a future work.
>
> # Comparison to different prunings
> - (SNIP vs. random pruning) We have tested random pruning for all models used in the paper for the same extreme sparsity levels. We also checked for a few relaxed sparsity levels (e.g. 70%). As expected, none of the randomly pruned sparse models is able to learn properly (the loss does not decrease). All of them record accuracies around 10%, which is the case of random guessing for the 10-way classification task. This implies that the randomly pruned sparse network does not have enough capacity to learn to perform the task. One potential reason would be that random pruning does not ensure the basic connectivity in the network, which can hinder the flow of activations in the forward pass as well as the gradients in the backward pass. In the worst case, all connections between two layers can be pruned away resulting in a completely disconnected network.
> - (SNIP vs. magnitude based pruning) We have also tested the pruning based on the magnitude of the initial weights and weights updated for a few iterations. We ensured to use the variance scaling initialization as the same as SNIP. As a result, the magnitude based pruning achieves the accuracies that are lower than the results with SNIP (e.g. 17.7% (Magnitude) vs. 14.99% (SNIP) on Alexnet-s).
>
> # Comparison to distillation
> - The objective of knowledge distillation is to transfer knowledge from the teacher network to the student network. Typically, this is achieved by enforcing the student network outputs the same as the teacher network (e.g. matching output activations or Jacobians). Hence, in order to perform knowledge distillation, a practitioner needs to pre-train the teacher network, and importantly, design the student network (smaller than teacher) in advance. Therefore, knowledge distillation can be complementary to SNIP; SNIP can be used to find the student network which is then trained with the objective of knowledge distillation.
>
> # Pruning a large architecture for architecture search
> - In fact, we have conducted experiments on a bulky architecture (by densely connecting residual blocks in ResNets) and applied SNIP to prune connections. As a preliminary result, we found out that the obtained architecture turned out to be somewhat different from the original ResNets, and yet improves the performance (1-2% increases in several variants of ResNets). We believe that this is an interesting direction to pursue, and we are planning to investigate more.
>
> # (Additional) Tiny-Imagenet results
> - Additionally, we have conducted more experiments on the Tiny-Imagenet classification task. Tiny-Imagenet is much larger and more complex than CIFAR-10, however, we observed that SNIP was still able to prune a large amount of parameters while achieving a comparable accuracies to the reference network. Please check the results in Table 4, Appendix C.
>
> We hope our response addresses the reviewer’s comments adequately. Otherwise, please leave us any further comments - we will do our best to update further.

---

### Official Review · AnonReviewer1 · 2018-11-02
**Thorough experimental evaluation of a simple method to prune neural networks before training. The same idea seems to have been already proposed in the literature.**

**Rating:** 7
**Confidence:** 4

**Review:**

This work introduces SNIP, a simple way to prune neural network weights before training according to a specific criterion. SNIP identifies prunable weights by the normalised gradient of the loss w.r.t. an implicit multiplicative factor “c” on the weights, denoted as the “sensitivity”. Essentially, this criterion takes two factors into account when determining the relevance of each weight; the scale of the gradient and the scale of the actual weight. The authors then rank the weights according to their sensitivity and remove the ones that are not in the top-k. They then proceed to train the surviving weights as normal on the task at hand. In experiments they show that this method can offer competitive results while being much simpler to implement than other methods in the literature.

This paper is well written and explains the main idea in a clear and effective manner. The method seems to offer a viable tradeoff between simplicity of implementation and effective sparse models. The experiments done are also extensive, as they cover a broad range of tasks: MNIST / CIFAR 10 classification with various architectures, ablation studies on the effects of different initialisations, visualisations of the pruning patterns and exploration of regularisation effects on a task involving fitting random labels.

However, this work has also an, I believe important, omission w.r.t. prior work. The idea of using that particular gradient as a guide to selecting which parameters to prune is actually not new and has been previously proposed at [1]. The authors of [1] considered unit pruning but the modification for weight pruning is trivial. It is worth pointing out that [1] is also discussed in one of the other citations of this work, namely [2]. For this reason, I believe that the main contribution of this paper is more on the thorough experimental evaluation of an existing idea rather than the proposed sensitivity metric.


As for other general comments:

- The authors argue that SNIP can offer training time speedups by only optimising the remaining parameters. In this spirit, the authors might also want to discuss about other works that seem relevant to this task, e.g.  [3, 4]. They also allow for pruned and sparse networks during training (thus speeding it up), without needing to conform to a specific sparsity pattern.

- SNIP seems to be a good candidate for applying it to randomly initialised networks; nevertheless, a lot of times we are also interested in pruning pre-trained networks. Given that SNIP is relying on the magnitude of the gradient to determine relevance, how good does it handle this particular case (given that the magnitude of the gradients is close to zero at convergence)?

- Why is the normalisation of the magnitude of the gradients necessary? The normalisation doesn’t change the relative ordering so we could simply just rank according to |g_j(w; D)|.

- While the experiment at section 5.6 is interesting, the result is still dependent on the a-priori chosen cut-off point “k”. For this reason it might be worthwhile to plot the behaviour of the network as a function of “k”. Furthermore, the authors should also refer to [5] as they originally did the same experiment and showed that they can obtain the same behaviour without any hyper parameters.

[1] Skeletonization: A Technique for Trimming the Fat from a Network via Relevance Assessment.
[2] A Simple Procedure for Pruning Back-Propagation Trained Neural Networks.
[3] Learning Sparse Neural Networks through L_0 Regularization.
[4] Generalized Dropout.
[5] Variational Dropout Sparsifies Deep Neural Networks.

---

> ### Author Response · Authors · 2018-11-07
> **SNIP idea is not the same as [1], and [1] has been cited already in the submission.**
>
> Thank you for the positive and constructive feedback. We appreciate that the reviewer finds that SNIP is clearly explained, viable and thoroughly evaluated.
>
> In this reply, we clarify the reviewer’s conjecture about the similarity between SNIP and the early work [1] (Skeletonization). Meanwhile, responses to the other comments will be provided in a succeeding reply.
>
> # Summary
> - It is incorrect to conclude that the idea behind SNIP is the same as the one presented in [1]. The differences are as follows.
>
> # SNIP vs. Skeletonization [1]
> - The fundamental idea behind [1] (also [2], OBD and OBS) is to identify elements (e.g. neurons, weights) that least degrade the performance when removed. Specifically, the saliency criterion in [1] is defined as $-dL/d\alpha$ (note the sign), which prunes elements that least increase the loss when removed. This means that this criterion, in fact, depends on the loss value before pruning, hence it requires the network to be pre-trained. Furthermore, to ensure minimal loss in performance, an iterative pruning scheme is employed in [1], leading to expensive prune -- retrain cycles.
>
> - In contrast, the saliency criterion in SNIP ($|dL/dc|$) is designed to measure the “sensitivity”, defined as how much influence an element has on the loss function regardless of whether it is positive or negative. This criterion alleviates the dependency on the value of the loss, thereby eliminating the need for pre-training. This is a fundamental conceptual difference of our approach. Consequently, the network can be pruned at single-shot prior to training. Moreover, we would like to point out that this aspect of SNIP allows us to interpret the retained connections (Section 5.4). Notice, such an experiment is not plausible (if not impossible) in previous works including [1].
>
> - Furthermore, in [1], robust auxiliary loss function ($L_1$) and exponentially decaying moving average (within the learning process) are required to suppress noise in the saliency score which is not the case in SNIP.
>
> - These conceptual and significant differences in the saliency criterion between SNIP and [1] result in fundamentally different pruning algorithms.
>
> # Citation of [1]
> - We would like to point out that we did not omit [1] and have cited [1] already in our submission (Sections 1 and 2).

---

> > ### Comment · AnonReviewer1 · 2018-11-12
> > **Similarity is important, further discussion in the submission is appropriate**
> >
> > Thank you for the detailed response on the differences between the two methods. Judging from your response, the main difference between the metric of [1] and SNIP is that SNIP considers the absolute value of the same gradient. This similarity seems important, and for this reason I believe that it is worthwhile to include this particular discussion in the main submission, especially when introducing the metric at section 4.1. Furthermore, I believe that the details about the way that each respective criterion is employed in practice (e.g. single-shot vs prune-retrain cycles, moving average of the metric etc.) are orthogonal to this discussion, as these concern specific choices rather than the core metric idea.
> >
> > I will wait for the authors to address all of the other points before I update my score.

---

> > > ### Author Response · Authors · 2018-11-21
> > > **Response to remaining comments**
> > >
> > > Thank you for the positive and constructive feedback. We recognize the importance of the suggested point on the difference between [1] and SNIP. We have added this discussion at the end of Section 4.1.
> > >
> > > We address the remaining comments below.
> > >
> > > # Training time speedup
> > > - We thank the reviewer for the nice pointer. We find that [3] reports the expected FLOPs ([4]: a rough observation) which is essentially attributed to the sparsity level. Notice, however, the maximum speedup by [3, 4] is achievable at the end of sparsification because they reach their maximum sparsity at the end of the process (which is the case for most pruning algorithms). In contrast, SNIP starts with the maximum sparsity from the beginning. This means that the speedup that can be achieved by [3, 4] will be upper-bounded by SNIP. We have updated the paper with this at the end of Section 5.2.
> > >
> > > # SNIP on pretrained networks
> > > - We have tested SNIP on pretrained networks and observed that SNIP also achieves comparable accuracies on pretrained networks (e.g. errors on LeNets: 3.1% (pretrain) vs. 2.4% (no-pretrain) on LeNet-300-100; 1.2% (pretrain) vs. 1.1% (no-pretrain) on LeNet-5-Caffe). We believe that it is most likely due to the fact that the gradients are hardly exactly zero in practice.
> > >
> > > # Normalization
> > > - We add the normalization as a means to handling the moving average over multiple mini-batches, which may arise in the case of a large model or dataset as noted in Section 4.2.
> > >
> > > # Fitting random labels
> > > - We have conducted the experiment with varying sparsity levels ($\kappa = 10, 30, 50, 70, 90, 99$) and updated the paper with this result in Figure 7, Appendix B. We have also cited [5] in Section 5.6.
> > >
> > > # (Additional) Tiny-Imagenet results
> > > - Additionally, we have conducted more experiments on the Tiny-Imagenet classification task. Tiny-Imagenet is much larger and more complex than CIFAR-10, however, we observed that SNIP was still able to prune a large amount of parameters while achieving a comparable accuracies to the reference network. Please check the results in Table 4, Appendix C.
> > >
> > > We hope our response addresses the reviewer’s remaining comments adequately. Otherwise, please leave us any further comments - we will do our best to update further.

---

> > > > ### Comment · AnonReviewer1 · 2018-11-23
> > > > **Response to the rebuttal**
> > > >
> > > > Thank for addressing all of my comments.
> > > >
> > > > - It is interesting to see that the error of the network is higher when pruning a pre-trained model. This seems to suggest that SNIP might be less effective on pre-trained networks. Of course, this conclusion could also be specific to the toy MNIST task, so further investigation is necessary to verify if this is indeed the case.
> > > >
> > > > - It seems that the explanation for [1] in the updated manuscript is not entirely accurate; it is stated that [1] uses - dl / d\alpha, with \alpha being a neuron. From what I understand, [1] instead prunes according to -dl / dc, where c is a 0-1 multiplicative term for a neuron (similar to what you introduced for weights), i.e. output_neuron_j = f(sum_i w_ij * c_i * input_neuron_i).

---

> > > > > ### Author Response · Authors · 2018-11-26
> > > > > **Thank you for the further comments.**
> > > > >
> > > > > Following the reviewer’s suggestion, we have made the explanation of [1] clearer and edited the manuscript as follows:
> > > > > (before) “$\alpha$ refers to neurons”
> > > > > (after) “$\alpha$ refers to the connectivity of neurons”
> > > > >
> > > > > We thank the reviewer for the constructive feedback, and we believe it indeed helped us improve the quality of the paper.

---

> > > > > > ### Comment · AnonReviewer1 · 2018-11-27
> > > > > > **Revised the score**
> > > > > >
> > > > > > Thank you for clarifying this further in the paper and for addressing all of my other comments. I updated my score to reflect this.

---

### Official Review · AnonReviewer2 · 2018-11-12
**Intriguing method to discover salient weights in an untrained neural network**

**Rating:** 8
**Confidence:** 5

**Review:**

Post rebuttal update/comment:

I thank the authors for the revision and have updated the score (twice!)

One genuinely perplexing result to me is that the method behaves better than random pruning, yet after selecting the salient neurons the weights can be reinitialized, as per the rebuttal:

> # Initialization procedure
- It is correct that the weights used to train the pruned model are possibly different from the ones used to compute the connection sensitivity. Given (variance scaled) initial weights, SNIP finds the architecturally important parameters in the network, then the pruned network is established and trained in the standard way.

First, there is work which states quite the opposite (e.g. https://arxiv.org/abs/1803.03635). Please relate to it.

Fundamentally, if you decouple weight pruning from initialization it also means that:
- the first layer will be pruned out of connections to constant pixels (which is seen in the visualizations), this remains meaningful even after a reinitialization
- the second and higher layers will be pruned somewhat randomly - even if the connections pruned were meaningful with the original weights, after the reinitialization the functions computed by the neurons in lower layers will be different, and have no relation to pruned weights. Thus the pruning will be essentially random (though possibly from a very specific random distribution). In other words - then neurons in a fully connected layer can be freely swapped, each neuron in the next layer behaves on al of them anyway we are thinking here about the uninitialized neurons, with each of them having a distribution over weights and not a particular set of sampled weights, this is valid because we will reinitialize the neurons). Because of that, I wouldn't call any particular weight/connection architecturally important and find it strange that such weights are found.

I find this behavior really perplexing, but I trust that your experiments are correct. however, please, if you have the time, verify it.

Original review:

The paper presents an intriguing result in which a salient, small subset of weights can be selected even in untrained networks given sensible initialization defaults are used. This result is surprising - the usual network pruning procedure assumed that a network is pretrained, and only then important connections are removed.

The contributions of the paper are two-fold:
1) it reintroduces a multiplicative sensitivity measure similar to the Breiman garotte
2) and shows which other design choices are needed to make it work on untrained networks, which is surprising.

While the main idea of the paper is clear and easy to intuitively understand, the details are not. My main concern is that paper differentiates between weights and connections (both terms are introduced on page iv to differentiate from earlier work). However, it is not clear what are the authors referring to:
- a conv layer has many repeated applications of the same weight. Am I correct to assume that a conv layer has many more connections, than weights? Furthermore, are the dramatic sparsities demonstrated over connections counted in this manner? This is important - on MNIST each digit has a constant zero border, all connections to the border are not needed and can be trivially removed (one can crop the images to remove them for similar results). Thus we can trivially remove connections, without removing weights.
- in paragraph 5.5 different weight initialization schemes are used for the purpose of saliency estimation, but the paragraph then says "Note that for training VS-X initialization is used in all the cases." Does it mean that first a set of random weights is sampled, then the sensitivities are computed, then a salient set of connections is established and the weights are REINITIALIZED from a distribution possibly different than the one used to compute the sensitivity? The fact that it works is very surprising and again suggests that the method identifies constant background pixels rather than important weights.
- on the other hand, if there is a one-to-one correspondence between connections and weights, then the differentiation from Karnin (1990) at the bottom of p. iv is misleading.

I would also be cautious about extrapolating results from MNIST to other vision datasets. MNIST has dark backgrounds. Let f(w,c) = 0*w*c. Trivially, df/dw = df/dc = 0. Thus the proposed sensitivity measure picks non-background pixels, which is also demonstrated in figure 2. However, this is a property of the dataset (which encodes background with 0) and not of the method! This should be further investigated - a quick check is to invert MNIST (make the images black-on-white, not white-on-black) and see if the method still works. Fashion MNIST behaves in a similar way. Thus the only non-trvial experiments are the ones on CIFAR10 (Table 2), but the majority of the analysis is conducted on white-on-black MNIST and Fashion-MNIST.

Finally, no experiment shows the benefit of introducing the variables "c", rather than using the gradient with respect to the weights. let f be the function computed by the network. Then:
- df/d(cw) is the gradient passed to the weights if the "c" variables were not introduced
- df/dw = df/d(cw) d(cw)/dw = df/d(cw) * c = df/d(cw)
- df/dc = df/d(cw) d(cw)/dc = df/d(cw) * w

Thus the proposed change seems to favor a combination of weight magnitude and the regular df/dw magnitude. I'd like to see how using the regular df/dw criterion would fare in single-shot pruning. In particular, I expect using the plain gradient to lead to similar selections to those in Figure 2, because for constant  pixels 0 = df/d(cw) = df/dc = df/dw.

Suggested corrections:
In related work (sec. 2) it is pointed that Hessian-based methods are unpractical due to the size od the Hessian. In fact OBD uses a diagonal approximation to the hessian, which is computed with complexity similar to the gradient, although it is typically not supported by deep learning toolkits. Please correct.

The description of weight initialization schemes should also be corrected (sec. 4.2). The sentence "Note that initializing neural networks is a random process, typically done using normal distribution with zero mean and a fixed variance." is wrong and artificially inflates the paper's contribution.  Variance normalizing schemes had been known since the nineties (see efficient backprop) and are the default in many toolkits, e.g. Pytorch uses the Kaiming rule which sets the standard deviation according to the fan-in: https://github.com/pytorch/pytorch/blob/master/torch/nn/modules/linear.py#L56.

Please enumerate the datasets (MNIST, Fashion-MNIST, CIFAR10) in the abstract, rather than saying "vision datasets", because MNIST in particular is not representative of vision datasets due to the constant zero padding, as explained before.

Missing references:
- Efficient Backprop http://yann.lecun.com/exdb/publis/pdf/lecun-98b.pdf discusses variance scaling initialization, and approximations to the hessian. Since both are mentioned in the text this should be cited as well.
- the Breiman non-negative garotte (https://www.jstor.org/stable/1269730) is a similar well-known technique in statistics


Finally, I liked the paper and wanted to give it a higher score, but reduced it because of the occurrence of many broad claims made in the paper, such as: 1) method works on MNIST => abstract claims it generally works on vision datasets 2) paper states "typically used is fixed variance init", but the popular toolkits (pytorch, keras) actually use the variance scaling one by default 3) the badly explained distinction between connection and weight and the relation that it implies to prior work. I will revise the score if these claims are corrected.

---

> ### Author Response · Authors · 2018-11-21
> **Response to R2 [2/2]**
>
>
>
> # Description of Hessian-based methods in Section 2
> - We agree that the complexity in computing the diagonal approximation of Hessian can be similar to that of the gradient.
> - We have updated the second line of the second paragraph (Modern advances) in Section 2 as follows:
> (before) "While Hessian based approaches suffer from the burden of the Hessian computation for large models,"
> (after) "While Hessian based approaches employ the diagonal approximation due to its computational overhead,"
>
> # Description of weight initialization in Section 4.2.
> - We find it correct that the idea of using variance scaled weight initialization is suggested in [Efficient Backprop; Section 4.6] and is also commonly employed in modern networks.
> - Therefore, we have updated the third paragraph in Section 4.2 as follows:
> (before) "..., typically done using normal distribution with zero mean and a fixed variance. However, even if the initial weights have a fixed variance, the signal passing through each layer no longer guarantees to have the same variance."
> (after) "..., typically done using normal distribution. However, if the initial weights have a fixed variance, the signal passing through each layer no longer guarantees to have the same variance, as noted by [Efficient Backprop]."
>
> # (Additional) Tiny-Imagenet results
> - Additionally, we have conducted more experiments on the Tiny-Imagenet classification task. Tiny-Imagenet is much larger and more complex than CIFAR-10, however, we observed that SNIP was still able to prune a large amount of parameters while achieving a comparable accuracies to the reference network. Please check the results in Table 4, Appendix C.
>
> # Dataset enumeration in the abstract (including results on Tiny-Imagenet)
> - We would like to ensure that we do not claim that our experimental finding on (Fashion-)MNIST will generalize to other "vision datasets". Therefore, we have updated the abstract to be more explicit as follows:
> (before) "... on image classification tasks …"
> (after) "... on the MNIST, CIFAR-10, and Tiny-Imagenet image classification tasks ..."
>
> # [Nonnegative Garotte by Breiman]
> - We recognize the relevance and have cited it in the beginning of Section 4.1.
>
> We hope our response addresses the reviewer’s comments adequately. Otherwise, please leave us any further comments - we will do our best to update further.

---

> > ### Author Response · Authors · 2018-12-04
> > **(Re-)initialization of parameters after pruning and before training**
> >
> > Thank you for the further comments. We have investigated [1] carefully and conducted experiments on (re-)initialization. Here we provide the results.
> >
> > We tested various models (LeNets, AlexNets, VGGs and WRNs) on MNIST and CIFAR-10 datasets for the same extreme sparsity levels used in our paper. As a result, we found that there are no differences in performance between re-initializing and NOT-initializing the subnetworks (after pruning by SNIP and before the start of training): 1) the final accuracies are almost the same (the difference is less than 0.1%) and 2) the training behavior (the training loss and validation accuracy curves) is very similar.
> >
> > This finding is contradictory to the one of the hypotheses in [1]: "When randomly reinitialized, a winning ticket learns more slowly and achieves lower test accuracy". However, their conclusions are based on magnitude based pruning (and there are differences in sparsity levels etc.), which might be the reason for the discrepancy.
> >
> > Notably, in the most updated version of [1] (27 Nov 2018), the authors explicitly state as a future work that "we intend to explore more efficient methods for finding winning tickets that will make it possible to study the lottery ticket hypothesis in more resource-intensive settings" or "... non-magnitude pruning methods (which could produce smaller winning tickets or find them earlier)".
> >
> > Being a single-shot pruning method at initialization, SNIP could be a method of choice for the further exploration of [1].
> >
> > References
> > [1] The Lottery Ticket Hypothesis: Finding Sparse, Trainable Neural Networks ( https://arxiv.org/abs/1803.03635 )

---

> > > ### Author Response · Authors · 2018-12-06
> > > **(cont'd) Response received from the authors of [1]**
> > >
> > > We asked the authors of [1] for clarification on the effect of (re-)initialization of subnetworks, and we received the following response.
> > >
> > > "The main statement of the lottery ticket hypothesis does not exclude the possibility that winning tickets are still trainable when reinitialized. Specifically, while the hypothesis conjectures that, given a dense network and its initialization, there exists a subnetwork that is still trainable with the original initializations, it does not require any particular behavior of this subnetwork under other initializations. Thank you for this comment; we will revise our language to make this clear."
> > >
> > > ( link to the full response: https://openreview.net/forum?id=rJl-b3RcF7&noteId=r1l-QxArJ4 )
> > >
> > > We hope this clears out the reviewer’s confusion on (re-)initializing subnetworks.

---

> > > > ### Comment · AnonReviewer2 · 2018-12-06
> > > > **Thanks!**
> > > >
> > > > Thank you for the extra experiments and clarifications, this makes the paper even more intriguing (score up).

---

> > > > ### Public Comment · (anonymous) · 2018-12-12
> > > > **Comparison of Lottery Ticket [1] and SNIP**
> > > >
> > > >
> > > >
> > > > We are the authors of the lottery ticket paper [1]. We have replicated the SNIP algorithm as presented in your paper in our own framework, and we performed several experiments that examine the relationship between SNIP and our paper. Our findings are:
> > > >
> > > > * Our winning tickets reach higher accuracy at higher levels of sparsity and learn faster than SNIP-pruned networks. (See https://openreview.net/forum?id=rJl-b3RcF7&noteId=S1xmvZRayE for details on the performance gap.)
> > > >
> > > > * SNIP-pruned networks can be randomly reinitialized as well as randomly rearranged (i.e., randomly choose the locations of unpruned connections within layers) with limited impact on their accuracy. However, these networks are neither as accurate nor learn as quickly as our winning tickets.
> > > >
> > > > The fact that SNIP-pruned networks can be rearranged suggests that SNIP largely identifies the proportions in which layers can be pruned such that the network is still able to learn, leaving significant opportunity to exploit the additional, initialization-sensitive understanding demonstrated by our results.
> > > >
> > > > We provide several graphs here (https://drive.google.com/drive/folders/1lpxJFpkF0Afq1rRqkEDnLcPN0kMV8BBC?usp=sharing) to support these claims.
> > > >
> > > > We are eager to hear your thoughts about our experiments!

---

> > > > > ### Author Response · Authors · 2018-12-12
> > > > > **It is misleading to compare Winning Lottery Ticket [1] against SNIP.**
> > > > >
> > > > >
> > > > > We believe that the comparison is misleading since [1] and SNIP focus on different (orthogonal) aspects of network pruning, and we elaborate this below.
> > > > > - SNIP focuses on finding a subnetwork at single-shot with a mini-batch of data, and shows that the subnetwork can be trained in the standard way. There are no hyperparameters involved in finding the subnetwork.
> > > > > - [1] states that there is a way to find a subnetwork -- based on costly iterative pruning and retraining (see the 2nd last paragraph of Section 2) -- and once this process finishes, the subnetwork can be trained as fast as training the original dense network.
> > > > > - In other words, SNIP is efficient in finding the subnetwork, and [1] is efficient in training the subnetwork.
> > > > > - Essentially, these two works are orthogonal exploring different aspects of network pruning, and therefore, it is misleading to compare these two approaches only on the aspect of training the subnetwork (i.e. provided figures).
> > > > >
> > > > > It is surely interesting to see how subnetworks obtained by different approaches compare to each other, and we hope to examine further into this once the code is released.

---

> ### Author Response · Authors · 2018-11-21
> **Response to R2 [1/2]**
>
> Thank you for the interest and positive feedback. We address the reviewer’s comments below.
>
> # (Fashion-)MNIST
> - First of all, both datasets are normalized before passing into the network, meaning that the dark region is actually not zero valued and the gradients from the dark region are not zero either.
> - Moreover, we ran the same experiment, but with inverted data as suggested by the reviewer (i.e. bright and dark regions are swapped). As a result, this led to the same results as in Figure 2. Please check the results from Figure 5, Appendix A.
> - Furthermore, we also ran the same experiment, but with $dL/dw$ as suggested by the reviewer. As a result, this led to the different results from Figure 2. In addition, using $dL/dw$ does not produce the same results when the dataset is inverted, as opposed to SNIP. Please check the results from Figure 6, Appendix A.
> - Therefore, it is incorrect to conclude that 1) "the proposed sensitivity measure picks non-background pixels", or 2) "the experiments on (Fashion-)MNIST is trivial and a property of the dataset, not of the method".
>
> # Connections (c) and weights (w)
> - We recognize that the initial description of connections (c) was not clear enough. Connections (c) are auxiliary indicator variables introduced to represent the connectivity of the parameters or weights (w). It is always one-to-one correspondence between connections (c) and weights (w) even for conv layers. Hence, the size of c and w are the same as m (i.e. sizeof(c) = sizeof(w) = m) as noted in Equation 3. Therefore, conv layers do not have more connections than weights, and the sparsity is measured correctly based on the total number of parameters (m).
> - We have updated the first line in the first paragraph of Section 4.1. as follows:
> (before) "we introduce auxiliary indicator variables c representing the presence of each connection."
> (after) "we introduce auxiliary indicator variables c representing the connectivity of parameters w."
>
> # Initialization procedure
> - It is correct that the weights used to train the pruned model are possibly different from the ones used to compute the connection sensitivity. Given (variance scaled) initial weights, SNIP finds the architecturally important parameters in the network, then the pruned network is established and trained in the standard way.
>
> # Differentiation from [Karnin 1990]
> - The fundamental idea behind [Karnin 1990] is to identify weights that least degrade the performance when removed. Specifically, the saliency criterion in [Karnin 1990] is defined as $-dL/dw$ (note the sign), which prunes weights that least increase the loss when removed. This means that this criterion, in fact, depends on the loss value before pruning, which requires the network to be pre-trained. Furthermore, to ensure minimal loss in performance, an iterative pruning scheme is employed in [Karnin 1990], leading to expensive prune -- retain cycles.
> - In contrast, the saliency criterion in SNIP ($|dL/dc|$) is designed to measure the “sensitivity”, defined as how much influence an element has on the loss function regardless of whether it is positive or negative. This criterion alleviates the dependency on the value of the loss, thereby eliminating the need for pre-training. This is a fundamental conceptual difference of our approach. Consequently, the network can be pruned at single-shot prior to training. This is in stark contrast to previous works including [Karnin 1990] where the saliency is measured using the entire dataset within an iterative optimization procedure.
> - These conceptual and significant differences in the saliency criterion between SNIP and [Karnin 1990] result in fundamentally different pruning algorithms.
> - We recognize the importance of this discussion and have added it at the end of Section 4.1.

---

### Public Comment · ~Thomas_Pfeil1 · 2018-10-09
**Comparison to random pruning**

Thank you for this interesting article. How does your method compare to random pruning using the same pruning rates?

---

> ### Author Response · Authors · 2018-10-09
> **Randomly pruned networks are unable to learn to perform the task.**
>
> Thank you for the interest in our work.
>
> We have tested random pruning for all models used in the paper for the same extreme sparsity levels. We also checked for a few relaxed sparsity levels (e.g. 70%).
>
> As expected, none of the randomly pruned sparse models is able to learn properly (the loss does not decrease). All of them record accuracies around 10%, which is the case of random guessing for the 10-way classification task.
>
> This implies that the randomly pruned sparse network does not have enough capacity to learn to perform the task. One potential reason would be that random pruning does not ensure the basic connectivity in the network, which can hinder the flow of activations in the forward pass as well as the gradients in the backward pass. In the worst case, all connections between two layers can be pruned away resulting in a completely disconnected network.

---

### Public Comment · (anonymous) · 2018-10-22
**Interpretability on More Complex Data**

Thanks for the interesting paper. On both MNIST and Fashion-MNIST, the object of interest is centered and the entire background of the image is black. Given you are using gradient information to select which connections should be removed, it seems obvious that the patterns you show in section 5.4 would occur.

Did you try this experiment on CIFAR? If so, would you be willing to share what you observed? It feels like this method could be duped by a dataset where the object of interest is not necessarily the brightest part of the image. More generally it feels like this technique would regress to the quality of random pruning on more complex datasets where the initial connection gradient is not informative.

---

> ### Author Response · Authors · 2018-10-23
> **Random pruning does not compare to SNIP, either for interpretation or for performing the task on any dataset.**
>
> Thank you for the interest in our work. We address your comments below.
>
> # Black background in (Fashion-)MNIST
> - Both datasets are normalized before passing into the network, meaning that the black region is not zero valued. Thus, gradients are not zero and do exist regardless of the intensity or region of the image.
> - Furthermore, we conducted the same experiment, but with reversed data (i.e. bright and dark regions are swapped), and this led to the same results as in Fig. 2: SNIP retains the same connections as it does with non-reverse data. This clearly indicates that there is no direct correlation between the image intensity (or brightness) and the connection sensitivity.
>
> # Visualization and interpretation of retained connections in the first layer (c_{l=1}) on CIFAR
> - The same experiment is not feasible with CIFAR, because the first layer in all tested networks is not fully connected. Hence, there is no one-to-one correspondence between the input and connectivity parameters c.
> - We can surely visualize convolutional parameters c, however, it will only reveal the level of sparsity rather than verifying the validity of the retained connections, which is not the purpose of this experiment.
>
> # Quality of random pruning against SNIP
> - In terms of interpretability, random pruning results in c that is completely random, on both (Fashion-)MNIST and CIFAR. Even though SNIP may not have the “reconstruction effect” on more complex data, it is likely to prune connections that are less important to perform the task - in case of image classification, it could be the background region. We believe that this is still far more meaningful than completely random patterns obtained by random pruning.
> - In terms of performance, random pruning fails to perform the task for all networks and datasets (see our response to Thomas Pfeil’s comment below), whereas SNIP achieves extremely sparse networks that are able to perform the task while maintaining the accuracy (see Table 2 for results on CIFAR).

---

### Public Comment · (anonymous) · 2018-10-26
**Details of Pruning Algorithm**

I'm unclear on the specifics of your pruning procedure. When you select weights to prune, do you:

1) Compute the sensitivities of all parameters globally (without considering which layer the parameters come from) and remove the k% of smallest-sensitivity parameters?

2) Compute the sensitivities of all parameters, normalize by layer, and then remove the k% of smallest-sensitivity parameters globally?

3) Remove the k% of smallest-sensitivity parameters in each layer?

4) Something else?

Thank you so much for the help!

---

> ### Author Response · Authors · 2018-10-26
> **The answer is 1.**
>
> Thank you for the question, and the answer is 1.
> The connection sensitivity is computed for all parameters globally and only top-k parameters are retained. Please refer to Equations (6) and (7) (also Lines 3 and 5 in Algorithm 1) where $m$ denotes the total number of parameters in the network.

---

### Public Comment · (anonymous) · 2018-10-28
**Relationship to Fisher pruning**

A very interesting paper!

I wanted to better understand the connection between SNIP and Fisher pruning (as described in abs/1801.05787).

Specifically, Fisher pruning would repeatedly: (1) train the network for some time and (2) remove the least important parameter. Let's say we modify Fisher pruning as follows:

* reduce the amount of training to just a single batch;
* remove all the parameters we want to remove at once (rather than one by one);
* when determining parameter importance, use absolute value instead of squared gradient;
* use variance-scaled initial weights.

How close to SNIP algorithm would this get us?

Also, another question: do you think SNIP will work if it's adapted to prune entire feature maps, i.e., channels (as discussed in the paper I quoted)? The rationale is that the CNN FLOPs cost is not affected much unless an entire channel is removed.

Thanks!

---

> ### Author Response · Authors · 2018-10-30
> **The modified Fisher pruning is different from SNIP and it is unclear whether it would lead to effective pruning.**
>
> Thank you for the interest in our work.
>
> # SNIP vs. modified Fisher pruning
> - The main idea behind Fisher or in general Hessian based pruning methods (e.g. OBS, OBD) is to remove parameters that least affect the loss at a local minimum based on second-order information. However, if we modify Fisher pruning as mentioned above, it does not satisfy the local minimum assumption or use second-order information. Therefore it is not clear whether this modified criterion would lead to effective pruning.
> - Furthermore, this resulting pruning criterion would be $|dL/dw|$ which is different from SNIP criterion ($|dL/dc|$) and does not measure the connection sensitivity as discussed at the end of Section 4.1.
>
> # SNIP for channel pruning
> - We believe that extending SNIP to channel pruning is surely feasible (e.g. by measuring connection sensitivities over channels). This can further save computational complexity and is an interesting direction to pursue for future work.

---

### Public Comment · ~Decebal_Constantin_Mocanu1 · 2018-11-20
**Related work - Sparse Evolutionary Training (SET)**

Dear authors,

Thank you very much for the very interesting work. Although it is omitting to cite our previous publications on this topic which transmit a similar message "neural networks shall and can have a sparse connectivity before training at no loss in accuracy", your paper is a nice read and propose a nice method. Thus, I would kindly ask you to discuss in your paper the relation between SNIP and our two previous publications on this topic. If you are not familiar with our work, to help with this discussion below are the main findings of our papers:

1) Mocanu et al.: “A topological insight into restricted Boltzmann machines”, Machine Learning, 2016 ( https://link.springer.com/article/10.1007%2Fs10994-016-5570-z ) where we show that we can create sparse connected Restricted Boltzmann Machines before training using some data statistics. These sparse RBMs can achieve similar performance with their fully connected counterpart or with the ones that use prune – retrain cycles procedures.

2) Mocanu et al.: “Scalable Training of Artificial Neural Networks with Adaptive Sparse Connectivity inspired by Network Science”, Nature Communications, 2018 ( https://www.nature.com/articles/s41467-018-04316-3 ). The original submitted version of this paper has been posted on arXiv from July 2017 (https://arxiv.org/pdf/1707.04780v1.pdf). On short, it says that the number of parameters in the deep neural networks fully connected layers can be quadratically reduced before training using our proposed method, i.e. Sparse Evolutionary Training (SET). SET starts from a Erdős–Rényi random graphs connectivity and further on it uses an evolutionary process during training to adapt the sparse connectivity to the data. In this way, SET is able to achieve with few order of magnitude faster training time and quadratically lower memory footprint for sparse deep nets in all stages (e.g. design, training, inference). We tried SET in combination with three neural network models, i.e. RBMs, MLPs, CNNs, on 15 datasets and our results show that the sparse models trained with SET achieve usually better (or at least the same) accuracy with their fully connected counterparts.

For any reason, the code of our proposed method is available online:
https://github.com/dcmocanu/sparse-evolutionary-artificial-neural-networks

To summarize, in my opinion, one important difference between our works is that you still have a pruning step before training, while we start directly with a sparse connected network.

I am looking forward to hear your opinion.

Best wishes,
Decebal

---

### Public Comment · ~Vaishnavh_Nagarajan3 · 2019-02-01
**SNIP on low-dimensional data?**

I've some questions relating to R2's comments based on a different intuition. First of all, I agree with R2 that it seems quite surprising and interesting that the original initialization doesn't matter after pruning the connectivities. However, I've a subtle disagreement with their intuition that "neurons in a fully connected layer can be freely swapped" and that this makes SNIP seem somewhat like random pruning. I don't think the neurons are necessarily swappable after removing the initialized weights. This is because the neurons in the first layer may be connected to different unique subsets of pixels, thus making the first layer neurons of the pruned network distinct from each other. Since the first layer neurons are hence unique, the neurons in the second layer, which may similarly be connected to unique subsets of first layer neurons, would also become distinct from each other. Extending this logic across all layers, it thus seems plausible that the connectivities across all layers have much more meaning to them than random pruning.

However there are situations where my intuition seems to weaken, and I'm curious to know what the authors think about these cases. Consider a case where the underlying data is low-dimensional e.g., 2D data, or high-dimensional data where only 2 dimensions matter. How does SNIP with reinitialization fare in this case, when compared to a) random pruning and to b) SNIP without reinitialization? In this case, it seems to me that SNIP pruning would connect all the first layer neurons to exactly the two relevant dimensions, thus making all those neurons fully equivalent to each other after pruning and after erasing the original initialization. Then, my intuition suggests that all subsequent layers are somewhat closer to being randomly pruned, and SNIP *probably* doesn't do anything interesting there.

To test this further, I wonder what happens if we do the following: create a network where the first layer connectivity is borrowed from a SNIP pruned network, and the connectivity in the remaining layers are based on random pruning; the network is then reinitialized. Does this network do worse than SNIP with reinitialization? If it doesn't, it will establish that SNIP basically relies on finding the right combinations of pixels that matter for learning, and the connectivity it learns in the subsequent layers doesn't really matter. If this network does worse, then it means that SNIP does in fact learn some interesting connectivities across all layers of the network, and it's not just about what combinations of pixels are relevant.

---

### Public Comment · ~Milad_Alizadeh1 · 2019-02-07
**Unofficial code on Github**

I have an unofficial PyTorch implementation of this on my Github if anyone's interested:

https://github.com/mi-lad/snip

---

### Public Comment · ~Szymon_Jakub_Mikler1 · 2019-02-08
**ICLR Reproducibility Challenge**

We tried to reproduce and examine SNIP pruning method. We weren't able to achieve as good results as the authors did on the LeNet-5 network. We weren't able to determine the reason yet. But still we found the method very interesing, more effective than any other one-shot pruning and certainly worth examining. We performed a few experiments about what connections are being pruned when using SNIP. We examine that separetely in three different scales: for whole network, for layers and for single neurons. Also, we wanted to determine if SNIP is a good approximation of what it ought to approximate (the difference in loss value caused by weight changing to 0). One might find all these experiments and sample replication code (PyTorch) under the following link:

https://github.com/gahaalt/ICLR-SNIP-pruning

---

### Public Comment · (anonymous) · 2019-06-11
**What about bias terms, should I parameterise them as b_mask * b_weight, and prune**

Does it make any difference

---

### Meta-Review · Area_Chair1 · 2018-12-16
**new method with thin evaluation**

**Confidence:** 5
**Recommendation:** Accept (Poster)

**Metareview:**

This method proposes a criterion (SNIP) to prune neural networks before training.  The pro is that SNIP can find the architecturally important parameters in the network without full training. The con is that SNIP only evaluated on small datasets (mnist, cifar, tiny-imagenet) and it's uncertain if the same heuristic works on large-scale dataset. Small datasets can always achieve high pruning ratio, so evaluation on ImageNet is quite important for pruning work. The reviewers have consensus on accept. The authors are recommended to compare with previous work [1][2] to make the paper more convincing.

[1] Song Han, Jeff Pool, John Tran, and William Dally. Learning both weights and connections for efficient neural network. NIPS, 2015.

[2] Yiwen Guo, Anbang Yao, and Yurong Chen. Dynamic network surgery for efficient dnns. NIPS, 2016.